# The Role of Physiotherapy in the New Treatment Landscape for Haemophilia

**DOI:** 10.3390/jcm10132822

**Published:** 2021-06-26

**Authors:** Sébastien Lobet, Merel Timmer, Christoph Königs, David Stephensen, Paul McLaughlin, Gaetan Duport, Cédric Hermans, Maria Elisa Mancuso

**Affiliations:** 1Hemostasis and Thrombosis Unit, Division of Hematology, Cliniques Universitaires Saint-Luc, Avenue Hippocrate 10, 1200 Brussels, Belgium; cedric.hermans@uclouvain.be; 2Neuromusculoskeletal Lab (NMSK), Institut de Recherche Expérimentale et Clinique, Université Catholique de Louvain, Avenue Mounier 53, 1200 Brussels, Belgium; 3Secteur de Kinésithérapie, Cliniques Universitaires Saint-Luc, Avenue Hippocrate 10, 1200 Brussels, Belgium; 4Van Creveldkliniek, University Medical Center Utrecht, Utrecht University, 3584 Utrecht, The Netherlands; M.A.Timmer@umcutrecht.nl; 5Department of Paediatrics and Adolescent Medicine, Haemophilia Treatment Centre, Goethe University, 60323 Frankfurt, Germany; christoph.koenigs@kgu.de; 6Kent Haemophilia Centre, East Kent Hospitals University NHS Trust, Canterbury CT1 3NG, UK; david.stephensen@nhs.net; 7Haemophilia Centre, Royal London Hospital, Barts Health NHS Trust, London E1 1BB, UK; 8Katharine Dormandy Haemophilia Centre and Thrombosis Unit, Royal Free London NHS Foundation Trust, London NW3 2QG, UK; p.mclaughlin@nhs.net; 9Association Française des Hemophiles, 75739 Paris, France; gaetan.duport@afh.asso.fr; 10Center for Thrombosis and Hemorrhagic Diseases, IRCCS Humanitas Research Hospital, 20089 Rozzano, Italy; mariaelisa.mancuso@humanitas.it

**Keywords:** physiotherapy, haemophilia, physiotherapy, EHL concentrates, emicizumab, trough levels, gene therapy

## Abstract

The physiotherapist plays an essential role for people with haemophilia, an inherited bleeding disease responsible for musculoskeletal complications. Yet, with the advent of new and advanced therapies, the medical landscape is changing, and physiotherapy must adapt alongside. This paper considers whether there will still be a need for physiotherapy in the era of advanced therapies, and discusses ways in which services should evolve to complement emerging treatment paradigms for haemostasis in people with haemophilia. Ultimately, physiotherapy will remain an important element of care, even for people with little joint damage and low risks in the era of the new mild phenotype. However, competencies will need to evolve, and physiotherapists in both primary care and specialist treatment centres should work with haematology colleagues to develop more sensitive tools for detecting early joint changes. Physiotherapists will also play a crucial role in counselling and physically coaching, monitoring the musculoskeletal status of people with haemophilia who have transitioned to new treatments.

## 1. Introduction

In the absence of treatment, joint and muscle bleeds are common in people with haemophilia (PWH) [1], and progression to arthropathy is often unavoidable [2,3]. Haemarthrosis leads to synovial hypertrophy and neoangiogenesis, further increasing susceptibility to mechanical damage [4]. The inflamed synovium affects the cartilage, in turn releasing cytokines and metalloproteinases and inducing chondrocyte apoptosis [4]. The joints most commonly affected are the elbows, knees, and ankles [5,6,7,8]. Haematoma can also affect muscle strength, and may lead to shortening of the muscle due to fibrosis [9]. There is clear evidence of an association between chronic joint and muscle damage and worsening clinical and quality of life outcomes in people with haemophilia [10]. Much of haemophilia care focuses on pharmacological parameters to prevent bleeds, improving but not eliminating joint disease. Since long-term outcomes are mainly musculoskeletal [3,6,11], a comprehensive care model is needed to achieve better outcomes in PWH [12,13,14]. 

Three forms of the disease are defined based on the plasmatic residual levels of FVIII (haemophilia A) or FIX (haemophilia B): mild, moderate, and severe. The risk of bleeding and, consequently, the risk of joint damage is related to the severity of the disease, being very high in untreated severe haemophilia and much less relevant in mild haemophilia [14]. With respect to musculoskeletal outcomes, for decades, major efforts have been focused on severe haemophilia to prevent crippling arthropathy, and prophylaxis was first and widely used in this form of the disease. More recently, long-term follow-up data demonstrate that patients with moderate haemophilia may also develop joint damage and may need prophylaxis [15]. 

The aim of pharmacological therapy is to prevent bleeding and reduce subsequent progression to arthropathy. The mainstay of treatment is factor replacement injected or infused intravenously on a regular basis. Prophylaxis, which has been shown to improve quality of life [16,17], is the current recommended standard of care for people with severe haemophilia, and some with moderate disease [14]. However, prophylaxis does not give full and permanent preservation of joint health [18,19], and the joints of PWH remain different from the general population [20,21]. The last decade has seen the introduction and increasing use of extended half-life (EHL) replacement products, which allow for better protection and fewer infusions. EHL products, such as primary prophylaxis in children, have the potential to protect the joints [22,23], and as secondary or tertiary prophylaxis in adults may slow down progression of joint damage [24]. More recently, non-replacement treatment options have emerged, and emicizumab is already licensed for prophylaxis in haemophilia A [25]. Additional non-replacement approaches in development have various targets, including tissue factor pathway, antithrombin, and anti-activated protein C [26,27,28], and can be used also in haemophilia B patients. Furthermore, advanced therapies, such as ultra-long-acting agents or gene therapy, are in late-stage development for people with severe haemophilia. With the classical intravenous replacement therapies using clotting factor concentrates, peaks of activity after infusions and trough concentrations before the next infusion are obtained, while non-replacement therapies are delivered subcutaneously, have a prolonged effect, and allow a steady improvement of the global clotting activity. These advanced therapies are creating a generation termed the new mild: PWH who were born with severe haemophilia, but whose advanced treatment means their phenotype now falls into the mild range of bleeding. Although bleeds are less frequent, these new mild PWH may have worse bleeds when they do occur, and may be less well-equipped to deal with them. With this in mind, there is a clear need to offer preventative education, counselling, and support.

In partnership with pharmacologic options, non-pharmacological management is critical to preserve or improve joint health and physical functioning and to encourage regular physical activity [1,29]. Physiotherapy plays an important role in prevention of—and rehabilitation after—joint and musculoskeletal bleeds, and is a key component of comprehensive care for PWH [1,5,14]. Despite significant advances in pharmacological treatment, there remains insufficient inclusion or recognition of musculoskeletal involvement as an integral part of haemophilia care. 

This paper considers whether there will still be a need for physiotherapy in the era of advanced therapies, and discusses ways in which physiotherapy services should evolve to complement emerging treatment paradigms for haemostasis in PWH [11,12]. It is also intended for local primary-care physiotherapists, who must also be aware of the evolution of treatments in order to interact effectively with both PWH and the team in the haemophilia treatment centre.

## 2. Pharmacologic Innovation: Impact on the Musculoskeletal System 

In the last five years, replacement therapy has been enriched with the advent of FVIII/FIX molecules purposely modified with enhanced pharmacokinetic properties that deliver better protection over a prolonged period. The main difference between replacement and non-replacement therapy is the shape of the pharmacokinetic curve generated: replacement delivers peaks and troughs, while non-replacement therapy offers a stable steady state (Figure 1) [28]. The peaks generated with replacement therapies are titratable and usually reach levels within the normal range of coagulation activation, while the flat steady state curve achieved with emicizumab, a bispecific antibody mimicking the action of FVIII, generally corresponds to an unmodifiable coagulation activation similar to that displayed by people with mild haemophilia with no ability to correct FVIII equivalent activity in the normal range. Similarly, with gene therapy, there is no oscillation: the factor is produced in each individual at a constant level, giving a flat profile with no peaks and troughs, although the exact level achieved widely varies between individuals as for the level of the steady state generated by emicizumab. A further key difference between replacement and non-replacement products is that replacement can be adapted to suit different needs, such as bouts of intense physical activity or physiotherapy. Whether the stable steady state generated by emicizumab or gene therapy is enough to allow intense physical activity or physiotherapy without bleeding complications is still unknown.

Individual treatment choice is therefore governed by factor levels, as well as joint status, lifestyle, and preference for physical activity [30]. At the present, PWH using factor products, such as prophylaxis, may need to adapt physical activity to suit their factor levels, or vice versa [29], and there is some evidence that at least 40–65% normal factor levels are required to prevent bleeding during high-risk, vigorous, or contact activities, depending on the presence or not of pre-existing joint morbidity [31,32]. 

In the future, if advanced replacement therapies move patients to a mild phenotype, they will potentially have constant factor correction, and enjoy zero bleeds for most activities of daily living [28]. Constant levels over longer intervals may allow PWH to achieve the same activity levels as for the general population [33]. However, it is expected that advanced treatments, such as ultra-long-acting factor concentrates or gene therapy, will be accessible only to adults, who will likely already have some level of pre-existing arthropathy. At the present time, the impact of new therapies on established arthropathy or bone mineral density remains unknown. 

While new products have enriched treatment options, many patients still experience bleeds. It is known that patients with worse clinical joint assessment (assessed by joint synovitis) and more joint bleeds are more likely to experience joint damage [8]. Discontinuing early prophylaxis in severe haemophilia A has been observed to lead to deterioration of joint status despite low bleeding rates, highlighting the need to continue monitoring patients for early signs of arthropathy despite low or zero bleed rates [34]. We hope that innovations in treatment will enable PWH to be more physically active without the need for additional treatment. However, it remains to be seen whether advanced therapies will allow all levels of physical activity without additional protection or individually planned prophylaxis [32], and certainly over the coming decades, arthropathy will continue to be an issue for PWH. It is clear, therefore, that physiotherapy remains an important element of care even for patients with limited joint damage and lower risks.

## 3. Comprehensive Care: The Evolving Role of the Physiotherapist

Physiotherapy has adapted to many changes in the haemophilia landscape in recent decades (Figure 2), and is now recognised as an essential part of the multidisciplinary team [12,14,35,36]. The WFH recommends regular replacement therapy to reduce bleed frequency in combination with physiotherapy aimed at preserving muscle strength and functional ability in order to prevent and treat chronic haemophilic arthropathy [14]. Despite these recommendations for comprehensive care [36], there is not always a multidisciplinary approach, and even where there is, the routine inclusion of a physiotherapist or musculoskeletal expert is variable. A recent EAHAD survey also showed considerable heterogeneity in physiotherapy roles and responsibilities across Europe [36], and this is likely to be even more diverse on a global scale [2]. This is particularly true regarding access to and the type of physiotherapy provided, as well as the skill set and autonomy of physiotherapists to make independent assessment and treatment decisions [36]; the full set of physiotherapy competencies are not currently recognised or utilised in all haemophilia treatment centres [12]. 

As well as supporting the concept of expert physiotherapy as part of the core multidisciplinary team for PWH, we recommend that PWH have direct access to a local physiotherapist. While the expert within the treatment centre oversees that person’s musculoskeletal health, it is important to be able to connect with local services who can provide day-to-day care and emergency help when it is needed.

## 4. Physiotherapy: Core and Specific Roles in PWH Management

The World Congress for Physical Therapy states that physiotherapy aims to “develop, maintain and restore maximum movement and functional ability throughout the lifespan” [37]. The role of the physiotherapist is to work in partnership with the PWH to assess and find meaningful management strategies to prevent and mitigate the impact of disease on function and structure, activity limitations, and participation restrictions, and to support people in achieving and maintaining optimum functioning in interaction with their environment [38,39]. Personalised and individualised exercise medicine allows PWH to benefit from better correction of their factor deficit, while providing them with knowledge to make informed decisions about their life. How a physiotherapist delivers care varies between different people, their needs, fears, or beliefs, especially when considering that 50% of people with bleeding disorders have a fear of exercise-induced bleeding or pain [29]. Within the comprehensive care setting, a combined multidisciplinary clinic review should be used to develop an individualised medical and physical care plan for each person. With the advent of new therapy leading to almost zero bleeds and fewer visits to hospital, physiotherapists will be crucial to offer guidance and return to optimal function after bleeds and injuries, prevent rebleeds or synovitis, maintain pristine joint health, and support people with existing arthropathy to fully benefit from the quality of life improvements that newer treatments offer. 

Physiotherapy encompasses more than just the treatment of musculoskeletal deficits: it forms part of a wider biopsychosocial network, providing behavioural and cognitive counselling, support, and advocacy, underscored by an understanding that functioning and disability are consequences of the interaction between a person’s health and their physical and social environment [13,29]. Physiotherapists also have a role in the care of female haemophilia carriers—especially those who are symptomatic—and people with other inherited bleeding disorders, such as von Willebrand disease. This requires a deep empathetic understanding of each person’s life in order to provide individualised and informed care [2,13] in three complementary areas: education, monitoring, and treatment or prevention, and we will look at each in turn from the perspective of caring for PWH. 

### 4.1. Providing Education

The advice and support that may be provided for PWH varies by life stage. For children and adolescents, there is a strong desire to do normal activities and competitive sports like their peers, and here, physiotherapy assessment should be focused on physical literacy, physical competency, and developmental norms. The traditional role for PWH has been on the sidelines, but this is not the case today. The physiotherapist is well-placed to educate not only PWH, but also to offer support and advice to coaches and trainers to ensure the safety of PWH who choose to play competitive sports. Education for PWH therefore includes counselling for sports participation and selection, with consideration of which are appropriate choices, and even which roles within a sports team are less prone to injury. For adults, the interaction may be guided by occupational needs and ability, plus daily life activities, with physiotherapy geared to the specific functional requirements of a person’s job and life situation. Older adults may rely on the physiotherapist to remain independent in daily functioning, for advice on safe and independent mobility, and help balancing the potential risk of falls against a background of widespread joint disease and physical deconditioning. 

Education and empowerment with knowledge allows PWH and their families to detect injuries and take adequate precautions [2], and helps drive behavioural change by allowing PWH to be active contributors to their own health and wellbeing. This includes supporting PWH to manage themselves by advising, motivating, and supporting them to be physically active [9], and to best manage joint or musculoskeletal problems as independently as possible. This also includes enabling people to do activities and exercises that may be of value, and encouraging and advising on appropriate nutrition, which plays a vital role in preventing muscle wastage [9]. The physiotherapist must also work with PWH to promote good health literacy regarding their own physical condition, help develop an action plan for joint complaints, and teach PWH when to contact the haemophilia treatment centre for a suspected bleed. 

The physiotherapist has a key role in educating and supporting PWH. We argue this will be especially important in children treated with optimal primary prophylaxis or bispecific antibodies who are potentially not experiencing a traditional bleed frequency. In this context, the physiotherapist’s task will be to teach PWH how to self-evaluate joints on a regular basis, to quickly recognise the signs of potential bleeding, and to self-rehabilitate with appropriate exercises at home.

### 4.2. Monitoring Joint Health and Physical Functioning

Musculoskeletal status and physical functioning should be assessed regularly [2] to demonstrate treatment efficacy, and for early detection of joint change or functional limitations. This is best and most effectively done by a physiotherapist with expertise in haemophilia, and will continue to be the case as the pharmacological landscape changes. The WHO’s international classification of functioning, disability, and health (ICF) for measuring function in haemophilia prioritises capacities over difficulties, reinforcing the biopsychosocial model noted above [13,38,39]. The ICF framework recommends the use of outcome measures on all ICF levels (structure/function, activities, and participation) and combines patient-reported and performance-based outcomes. However, we predict that measurement of annual bleeding rate will no longer be a sensitive tool for assessing the effectiveness of haemostatic treatment, since it is likely that with advanced therapies, most patients will no longer bleed, or only infrequently [11]. Differentiating between bleeds and other painful episodes will become important, since those who experience fewer or minimal bleeds are often less skilled at managing them in a timely and appropriate manner. 

Point-of-care ultrasound can be considered for early detection of synovial changes, cartilage lesions and diagnosis of acute bleeding [6], although survey findings suggest that it is used routinely in only 60%, 43%, and 47% of haemophilia treatment centres in Europe, North America, and the rest of the world, respectively [40]. In addition, only a quarter of physiotherapists report being able to independently use the technology [36]. The introduction of pocket handheld ultrasound devices has the potential to rapidly expand the utility and reliability of this modality [41], and we predict this will increase usage statistics. However, in the future there will be a need for more sensitive musculoskeletal assessment tools based on function and movement [11], since it is reasonable to expect that PWH receiving advanced therapies will have greater physical ambitions, although in the short term the transitional population may have residual pain and functional impairment from pre-existing arthropathy. Surveillance efforts may combine high-level physical performance with the ability to perform day-to-day activities. Assessment approaches such as gait analysis, balance assessment, or strength deficit assessment may be useful to detect the first signs of joint disease, or for patient follow-up. A weakness of such assessment approaches is establishment of normative values, and knowing when to consider the presence of a deficit. Biochemical markers may also be useful—but although they are widely used in other destructive joint diseases they are lacking in haemophilia, and their development may be more complex than in systemic inflammatory arthritis [42]. We must also balance the burden of monitoring in a potentially less medicalised era of more efficacious treatments. 

The physiotherapist with a vocation for fundamental and clinical research will be the ideal person to refine or develop new evaluation tools, whether they are based on physical performance or self-evaluation questionnaires. These tools will of course have to demonstrate better clinimetric qualities such as reproducibility and sensitivity, as well as sensitivity to change (responsiveness).

New therapies have a profound impact on blood coagulation, which should allow PWH to engage more actively in physical activities. Counselling and coaching provided by the physiotherapist appear instrumental to guarantee a smooth physical transition that should be harmonised to the haemostatic transition. By contrast with the rapid change in haemostatic product, the physical transition should be smooth in order to progressively adapt the musculoskeletal system to a new range of physical activities.

### 4.3. Treatment and Prevention

Physiotherapists are expert in musculoskeletal treatment and assessment of disabilities, and able to address key domains of the ICF model [38,39]. The concept of functioning includes all body functions, structures, activities, and an individual’s participation in society—covering both mental and physical components [39]. Activity is understood as an individual’s capability to execute a task or action by an individual; in contrast, activity limitation is the difficulty experienced in a given domain at a given moment [39]. Participation describes involvement in life situations [39]. We know from the CHESS II data that haemophilia reduces both physical and social activity [43]. Whilst this is more pronounced in moderate and severe disease, there is still an impact in people with mild haemophilia, with over one-third of patients reporting limitations [43]. 

Improvements in coagulation correspond with greater possibility of joint protection, and understanding this individual variation is an important element to performing physical activity and physiotherapy safely. The role of the physiotherapist will depend on each person’s musculoskeletal status prior to receiving gene therapy or other advanced treatment, especially if their factor levels remain below normal after infusion of a gene therapy product. Preliminary studies show that factor levels after gene therapy have high intra-individual variation, and may change over time [44]. 

In the future, discussion between the physiotherapist and haematologist should include the mechanism and pharmacokinetics of new therapies [11], and their strengths and limitations. This will enable physiotherapists to evolve their competencies and tailor personalised care plans to complement pharmacological management, as well as to confidently advocate emerging treatments and describe any necessary transitions. 

An important role of the physiotherapist within the haemophilia treatment centre is to maintain contact with the local physiotherapist treating PWH. Since it is a rare disease, local physiotherapists have little knowledge and experience with haemophilia. However, many patients live too far from the haemophilia treatment centre to visit on a weekly basis. The physiotherapist within the haemophilia treatment centre can provide information, make a shared treatment plan and should be easily available for questions. 

## 5. Future Considerations

The COVID-19 pandemic has challenged the traditional model of care provision, creating opportunities for virtual consultations and telerehabilitation as well as more equitable access to educational content with congresses going online. Such changes need to be evaluated as to their long-term benefit for local services and the people who use them, as well as the possibilities of ‘meta-networks’ of shared clinical expertise for the musculoskeletal management of PWH. Online teaching, mentorship, and support may be a way of helping others advocate for services locally where there are none, and establish collaborative clinical and research efforts that benefit all.

## 6. Discussion: Impact of Treatment Innovation on Physiotherapy in Haemophilia

Innovations invariably change the landscape of a disease. Physical activity, exercise, diet, and nutrition will remain key aspects of haemophilia management [9], but although there is increasingly a more positive attitude towards sports and activities in many PWH [7], their impact is often underestimated. Physical activity is important for everybody [33]; but for PWH appropriate activity is particularly important for recovery after musculoskeletal bleeds, and for managing established arthropathy [14]. A lack of physical activity can also be the root cause of many comorbidities, including obesity, cardiovascular disease, and osteoporosis [45,46]. Having a care model whereby the physiotherapist is able to offer education, support and treatment throughout the life course is critical to allaying fears about physical limitations, and encouraging healthy activity at every life stage.

The WFH have positioned physiotherapy as a life-long requirement for PWH [12], and we hereby emphasise the core role of the physiotherapist will not change with the advent of new innovations in pharmacological care for haemophilia. The presence of pre-existing muscle and joint damage from years of bleeding are still seen presently, and in the future a more preventative role will be key to optimize joint health outcomes. We advocate that comprehensive care means more than individual consultations with core team members in isolation, but that combined consultations of healthcare professionals are necessary so that optimum care for each individual PWH can be achieved. 

Treatment individualisation is important in haemophilia, as in all long-term chronic conditions [47]. Services should be coordinated to enable self-management and shared decision making, as well as to provide effective treatment and support [47]. Informed and shared decision making is important, and becomes increasingly so when there are more therapeutic options to choose from. As new pharmacological therapies evolve, we will need to review how routine assessments are done, and how people who have moved to a mild phenotype are monitored. As new surgeries evolve, so too will the role of the physiotherapist in rehabilitation—as well as potentially in helping to identify good candidates for certain orthopaedic interventions [Box 1].

Box 1Recommendations for principles of physiotherapy in the era of non-replacement therapies.
The physiotherapist is recognised as an essential member of the multidisciplinary team for the care of PWH.Physiotherapy will remain an important element of care even for people with little joint damage and low risks.The physiotherapist should be in contact with local practitioners who can provide day-to-day care outside the treatment centre. Experienced physiotherapists from expert haemophilia treatment centres should provide support, education, and resources for other providers. The physiotherapist will continue to play a key role in educating children and adults with haemophilia on how to identify and treat bleeds and how to undertake physical activity safely.Advanced therapies may widen the generational gap in the musculoskeletal com-plications seen in PWH; services will need to adapt to reflect that. Competencies will evolve, and physiotherapists should work with colleagues in haematology to develop more sensitive tools for the detection of early joint chang-es. Treatment interactions will also evolve; physiotherapists should seek ways to de-liver joint and injury assessments via telehealth platforms. Physiotherapy is critical to promote a harmonious haemostatic and physical state in PWH transitioned to new therapies.


Even where innovative therapies do move PWH from a severe to ‘new mild’ phenotype in terms of factor levels, we know that people with mild haemophilia can develop long-term joint damage and experience social and physical limitations [10,34,43]. In addition, people with a natural mild haemophilia show variation in their disease state due to other considerations such as inflammation or tissue damage. As such we need to continue to look after them. How often these people will be seen in regular clinical practice is an area that needs thought and planning, and we should be careful that the emerging era of non-replacement therapies does not underestimate the role of adjunct non-pharmacological care. Whilst there may be a case for blended care, and supporting PWH to perform tailored physical activities and exercises at home [48], physiotherapy adds value to the potential overall success of advanced therapies [12]. Some competencies will evolve, and the core role of the physiotherapist in performing a correct joint assessment and detecting changes early may require more sensitive tools [11]. In addition, the range of PWH that we see will widen, from older people who still present with arthropathy traditionally associated with severe haemophilia, to younger PWH who may need support to retain pristine joints but still require monitoring, with a varied spectrum in between. 

## 7. Conclusions

This group set out to ask whether there will still be a need for physiotherapy in the era of non-replacement therapies, and the answer is a resounding yes. We have shown that this will continue to be the case with the advent of novel therapies [33], regardless of factor levels or haemostasis achieved. Physiotherapy will remain crucial in PWH in the future, both in those treated with evolving therapies and also in those around the world currently not receiving adequate treatment. Ultimately, awareness of the importance of physiotherapy combined with an understanding of the evolution of haemophilia management will positively influence the future of care, and enable PWH to fully benefit from better treatment interventions. 

## Figures and Tables

**Figure 1 jcm-10-02822-f001:**
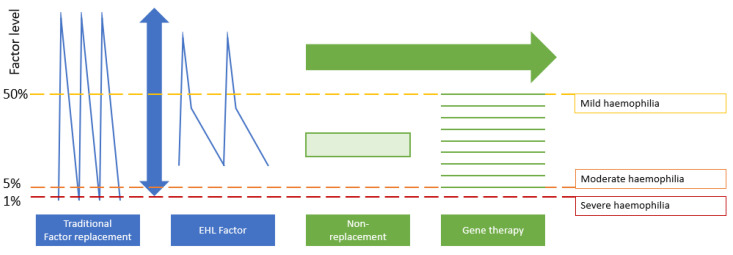
Key differences in haemostatic coverage with different pharmacological approaches for haemophilia [28]. Peak and trough patterns characteristic of traditional factor replacement and EHL factor are compared to the flat profiles of non-replacement and gene therapy. For gene therapy, the green lines represent nine theoretical patient profiles. These are shown to demonstrate the variation in levels that may be achieved, but that each person maintains a constant level over time. The figure is adapted and reproduced with permission from Elsevier Science and Technology Journals.

**Figure 2 jcm-10-02822-f002:**
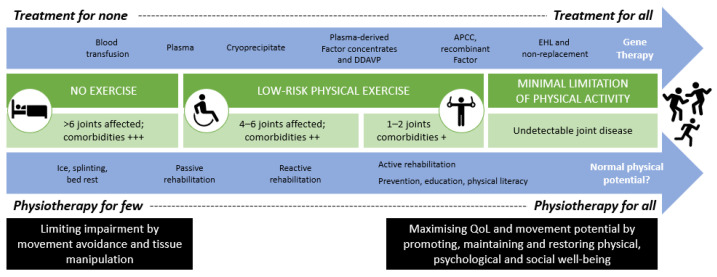
Timeline of physiotherapy adaptations within the haemophilia landscape. There has been a growing scope for physiotherapy and physical activity alongside increasingly effective medical management options. APCC, activated prothrombin complex concentrate; DDAVP, desmopressin; EHL, extended half-life products; QoL, quality of life.

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
