# Peer review of "The Role of Physiotherapy in the New Treatment Landscape for Haemophilia"

_jcm, 2021, doi:10.3390/jcm10132822_

Round 1
Reviewer 1 Report
I thank Lobet et al. for a very well written and much needed manuscript. I found the content to be well organized, the graphics to be very useful and well executed and the manuscript to be referenced well overall.
I kindly request the following revisions/additions are considered by the authors to help highlight the role of PT in HTCs and the care of patients with hemophilia and bleeding disorders:
- Nonfactor therapies/gene therapy are creating a generation of what many of us in the field refer to as the “new mild patient”; patients who were born severe and are now mild in their bleeding phenotype and effective clotting factor levels. These "new mild hemophilia patients" just like those born mild tend to have worse and more severe bleeding episodes, when they occur, secondary to delayed ability to identify and treat bleeding episodes, delayed factor administration resulting in increased factor consumption, prolonged recovery and a higher rate of complications. I kindly request that the authors make this point clear earlier in the manuscript to highlight the importance of preventative education and counseling by HTC provider and PT for these patients, who although may bleed less are still at risk for serious bleeding episodes.
- It would be worth mentioning in the manuscript that physiotherapists are not only valuable in the comprehensive care of patients with hemophilia but also hemophilia female carriers especially the low level symptomatic ones and other patients with inherited bleeding diathesis including von Willebrand disease and rare bleeding disorders cared for in hemophilia treatment centers (HTCs). Our therapists have also assisted patients with sickle cell disease and those who suffered a stroke as these patients are cared for at our HTC as well.
- Section 4.1 Providing education: “counseling for sports participation and selection” (which sports are safe for bleeding disorder patients, which positions within a sport are less prone for injury) is an exciting talk our young children/teens have with their PT. If possible clearly list that role as well as PT partnering with school gym class/physical education teachers and athletic trainers to ensure safety of PwH playing competitive sports and debunking myths that prevent the PwH from playing sports for fear of injuries.
- It is important in the manuscript to emphasize on adult PwH morbidities including obesity, cardiac disease, osteopenia/osteoporosis with fractures; many of which are attributed to decreased physical activity secondary to misconceptions and fears of injury when the care model does not include a physical therapist who educates, supports, evaluates and treats.
- POCMSKUS: I found the reported adoption rates to be low in this manuscript and the role minimalized. I recommend reviewing newer publications including “Use of ultrasound for assessment of musculoskeletal disease in persons with haemophilia: Results of an International Prophylaxis Study Group global survey by Danial M Ignas et al.”. POCMSKUS and handheld ultrasound probes are being incorporated more in the care of patients with hemophilia, drug trials and soon enough the technology will be available and affordable for patients and providers. Artificial intelligence modules that help detect bleeds are also being developed. Hence, partnering with PT’s on this tool will bridge the gap between HTCs and patients and prevent under/overtreatment of bleeding episodes or misdiagnosis. It is also another tool in the PT toolbox!
- The paper very nicely addresses the role of PT education, monitoring (perhaps worth mentioning joint assessment scores HJHS etc), treatment and prevention. Box 1 nicely summarizes recommendations for principles of PT in the era of non-replacement therapies. Despite clear rationale, I found the language the authors used to support the role to be unjustifiably timid:
BOX1: “PT APPEARS criticall--> change to PT IS critical”
Line 353: “The WFH have positioned physiotherapy as a life-long requirement for PWH [12], and we FEEL STRONGLY that the core role of the physiotherapist will not change with the advent of new innovations in pharmacological care for haemophilia – AT LEAST NOT IN THE SHORT- and mid-term when we will still see pre-existing muscle and joint damage from years of bleeding. We advocate that comprehensive care means more than individual consultations with core team members in isolation, but that combined consultations of healthcare professionals are necessary so that optimum care for each individual PWH can be achieved.”
STRONGLY FEEL--> change to EMPHASIZE/INSIST (I think feelings and beliefs don’t serve a good argument and there is enough data to substantiate PT role in this paper and others).
I recommend rewording as such Line 353: “The WFH have positioned physiotherapy as a life-long requirement for PWH [12], and we HEREBY EMPHASIZE that the core role of the physiotherapist will not change with the advent of new innovations in pharmacological care for haemophilia since pre-existing muscle and joint damage from years of bleeding are still seen presently and in the future a more preventative role will be key to optimized joint health outcomes. We advocate that comprehensive care means more than individual consultations with core team members in isolation, but that combined consultations of healthcare professionals are necessary so that optimum care for each individual PWH can be achieved.”
CONCLUSION: “We BELIEVE this will continue to be the case with the advent of novel therapies, regardless of factor levels or haemostasis achieved. Physiotherapy will remain crucial in PWH in the future, both in those treated with evolving therapies and also in those around the world currently not receiving adequate treatment.” Change BELIEVE to --> DEMONSTRATE, ARGUE, SHOW, EXHIBIT…etc.
Kindly look throughout the paper for other examples where minor rewording can help establish the emerging and ever evolving role of PT in the care of PwH.
- Lastly, I would recommend future considerations section is added to address the centers who read this paper and realize the role of PT but unfortunately don’t have access. This pandemic opened the doors for telehealth, online meetings, inter-departmental, institutional and international collaborations! Do the authors foresee a future where experienced HTC PT’s develop counseling/educational modules that are offered free for access for other providers to use with their patients (physical activity, sports participation, bleed recognition, RICE, rehabilitation, assistive devices…etc). Furthermore, it would be great if telehealth assessment of injuries is emphasized on as it is valuable and can help bridge the gap with patients who do not like to travel or do not have transportation to their HTCs. It also allows providers to partner with experienced national/international PT's to assess a patient and learn how to incorporate PT assessments as part of their exam or even better prove/advocate for the need for a PT after partnering with an expert PT. Something pre-pandemic would mean travel/honoraria whereas now is a zoom call!
No further comments. Thank you again.
Reviewer 2 Report
Dear Authors,
Congratulations for your paper that I think could contribute as an expert opinion in the field of hemophilia.
Page 2 line 69: please remove the point.
Page 4 line 148-149: please close the brackets.
Page 4 lines 165-173: I think that it could be better to underline the need of a multidisciplinary team where PT has a crucial role. The reader of your paper could have the message that the management of PWH is possible with an hematologist and a PT. Somewhere in the paper the term "physiotherapist" could be better replaced with "musculoskeletal expert".
Author Response
Thank you for the well-considered comments and suggestions. We have made our amends in tracked changes as requested, and note below our responses to each point.
Kind regards,
Sebastien Lobet
Page 2, line 69: thank you, this editorial amend has been made.
Page 4, line 148: thank you for picking up this typo!
Page 4, lines 165-173: we take your point, and have made amends (~line 171). 'PT or musculoskeletal expert' has been used for clarity.